# Responses of Protozoan Communities to Multiple Environmental Stresses (Warming, Eutrophication, and Pesticide Pollution)

**DOI:** 10.3390/ani14091293

**Published:** 2024-04-25

**Authors:** Guoqing Yuan, Yue Chen, Yulu Wang, Hanwen Zhang, Hongxia Wang, Mixue Jiang, Xiaonan Zhang, Yingchun Gong, Saibo Yuan

**Affiliations:** 1College of Fisheries and Life Science, Dalian Ocean University, Dalian 116023, China; yuanguoqing@ihb.ac.cn; 2State Key Laboratory of Freshwater Ecology and Biotechnology, Institute of Hydrobiology, Chinese Academy of Sciences, Wuhan 430072, China; chenjiaxiaoyue@foxmail.com (Y.C.); yuluwang@whu.edu.cn (Y.W.); zhanghw@ihb.ac.cn (H.Z.); hongxiawang@ihb.ac.cn (H.W.); jiangmixue22@mails.ucas.ac.cn (M.J.); zhangxiaonan23@mails.ucas.ac.cn (X.Z.); 3Ecological Environment Monitoring and Scientific Research Center, Ecology and Environment Supervision and Administration Bureau of Yangtze Valley, Ministry of Ecology and Environment of the People’s Republic of China, Wuhan 430014, China

**Keywords:** protozoa, functional group, global warming, eutrophication, pesticide pollution

## Abstract

**Simple Summary:**

Protozoa are the most abundant phagotrophic group in the biosphere and play an important ecological role in aquatic ecosystems. However, the effects of environmental stressors such as temperature changes, eutrophication, and pesticide pollution, on the protozoan communities in fresh waters remain poorly understood. The results of our study show that warming and eutrophication considerably promote an increase in protozoan biomass, and the combination of warming and pesticides can remarkably reduce the abundance, biomass and diversity of protozoa. Warming, eutrophication, and pesticide pollution affect protozoan diversity, community structure, and functional groups, either independently or interactively.

**Abstract:**

To explore the impacts of multiple environmental stressors on animal communities in aquatic ecosystems, we selected protozoa—a highly sensitive group of organisms—to assess the effect of environmental change. To conduct this simulation we conducted a three-factor, outdoor, mesocosm experiment from March to November 2021. Changes in the community structure and functional group composition of protozoan communities under the separate and combined effects of these three environmental stressors were investigated by warming and the addition of nitrogen, phosphorus, and pesticides. The results were as follows: (1) Both eutrophication and pesticides had a considerable promotional effect on the abundance and biomass of protozoa; the effect of warming was not considerable. When warming was combined with eutrophication and pesticides, there was a synergistic effect and antagonistic effect, respectively. (2) Eutrophication promoted α diversity of protozoa and affected their species richness and dominant species composition; the combination of warming and pesticides remarkably reduced the α diversity of protozoa. (3) Warming, eutrophication, and pesticides were important factors affecting the functional groups of protozoa. Interaction among different environmental factors could complicate changes in the aquatic ecological environment and its protozoan communities. Indeed, in the context of climate change, it might be more difficult to predict future trends in the protozoan community. Therefore, our results provide a scientific basis for the protection and restoration of shallow lake ecosystems; they also offer valuable insights in predicting changes in shallow lakes.

## 1. Introduction

Currently, shallow lake ecosystems are influenced by multiple environmental changes, among which global warming and water eutrophication are considered to be the most critical. Thus, these changes pose serious threats to these ecosystems. According to the United Nations Intergovernmental Panel on Climate Change (IPCC), the temperature in China’s mid-latitudes will rise by 3–4 °C by 2100 [1]. Warming leads to changes in the phenological rhythm of aquatic organisms and alters the nutritional relationships among species, thereby affecting the community structure of aquatic systems [2,3,4]. Concomitant with climate changes, the use of chemical fertilizers and the discharge of rural and urban sewage means lake eutrophication has become a serious cause for concern [5]; in combination, these stressors lead to the proliferation of phytoplankton and the degradation of zooplankton and aquatic plants and thereby drive an imbalance in the structure and function of aquatic ecosystems [6,7]. In addition, pesticides and other substances used in terrestrial agricultural production, which enter lakes through rainwater erosion or leaching, also pose a serious threat to lake ecosystems [8,9]. Previous research has shown that climate warming, nutrients, and pesticide pollution have significant individual and interactive effects on aquatic ecosystems and their food web [10,11]. However, little is known about the multiple environmental stresses on these tiny aquatic animals which are highly sensitive to changes in the lake ecosystem environment; exploring the impacts of multiple stressors of water eutrophication and pesticide pollution on the structure and biodiversity of those sensitive groups’ communities against the background of global change, will offer valuable insights to reveal the future trends of lake ecosystem changes and their protection and restoration.

Protozoa are a group of unicellular eukaryotes with diverse forms and wide distribution [12]; collectively they play important roles in aquatic ecosystems [13,14,15,16]. They are the most abundant phagotrophic groups in the biosphere and mainly feed on algae, bacteria, and organic debris, which to a certain extent, can affect the phytoplankton community structure and improve the water quality [17,18]. Moreover, protozoa are an important food source for higher aquatic animals such as fish and shrimp, and play an indispensable role in the material circulation and energy flow of the aquatic ecosystem [19,20,21]. In addition, because protozoa are highly sensitive to changes in the surrounding environment, they are often selected as indicators to monitor and evaluate the aquatic ecological environment. Thus, protozoa play an important role in aquatic ecological environmental monitoring and ecosystem protection and restoration [22,23,24].

Protozoa can be divided into five functional groups: algivores (A), bacterivores (B), algivores and bacterivores (A&B), predators (Pr; “R” was used for predator in the original reference [25]), and non-selective omnivores (N) [25,26]. Many researchers have studied the relationships between zooplankton functional groups and water quality in different aquatic ecosystems, including Chaohu Lake [27,28], Liangzi Lake [29], Hulun Lake [30], and Gahai Lake [31]. However, owing to the strong complexity and poor controllability of natural lake ecosystems, as well as the difficulty in identifying protozoa, the impacts and mechanisms of multiple environmental stressors such as eutrophication and pesticide pollution on the structure of protozoan communities in the context of global change remain unclear.

Mesocosm enclosure is a suitable method to simulate how changes in temperature, nutrients and other environmental factors affect the long-term dynamics of aquatic organisms in shallow lake ecosystems [32,33]. To date, this method has been widely used to study the responses of phytoplankton [34], zooplankton [35], zoobenthos [36], and aquatic plants [37] to environmental change.

The purpose of our study was to simulate shallow lake ecosystems under the stress of global warming and eutrophication by the addition of pesticides to a mesocosm experimental system and evaluating the combined effects of multiple environmental stressors on protozoan community structure and functional groups. Results from this research will be helpful in predicting future trends of changes in lake aquatic ecological environments. It also will provide a scientific basis for the protection and restoration of aquatic ecosystems in shallow lakes.

## 2. Materials and Methods

### 2.1. Experimental Design

All the experiments were conducted using a set of mesocosm simulation control experimental systems for shallow lakes. The setting of the mesocosm systems was the same as the previous study by our team [36]. The system comprised 48 cylindrical polyethylene water tanks (diameter: 1.5 m, depth: 1.4 m, volume: approximately 2500 L) and a control platform (Figure 1). The bottom of each mesocosm was filled with 10 cm of sediment which was collected from Lake Liangzi (30°11′3″ N, 114°37′59″ E) in the Yangtze River basin and homogenized and sieved through a 5 mm × 5 mm metal mesh to remove large debris, macrophyte seeds, and snails. After that, each mesocosm system was gradually filled to a depth of 1.2 m with about 2100 L aerated tap water, and then ten liters of original lake water was added to the mesocosm to inoculate plankton from nearby Lake Nanhu (30°28′57″ N, 114°22′34″ E). An aquarium pump was installed in each mesocosm to mix water. To stimulate the natural ecosystem in the mesocosms, turions of common submerged macrophytes were seeded in the sediment, and some fish, shrimps and snails were introduced in each tank evenly [36]. All the mesocosms were acclimated for two months before we began the experiments. The evaporation loss of the system was supplemented with deionized water and natural rainfall.

Three stressors were set in this study, including warming (W), eutrophication (E) and pesticide pollution (imidacloprid (P)). Two temperature levels (C: no addition for the control; W: continuous temperature rise of +3.5 °C), two nutrient concentration levels (C: no addition for the control; E: added nitrogen and phosphorus), and two pesticide concentration levels (C: no addition for the control; P: added imidacloprid) were designed in an orthogonal experiment. Eight treatments were established with six replicates per treatment, and all treatments were randomly assigned to the 48 water tank systems. The warming treatment was set as +3.5 °C higher than the control. The magnitude of the warming was based on model predictions from the historical meteorological data in the Yangtze River Basin, China [1]. Warming was achieved by using a computer-controlled system with temperature sensors (DS18B20; Maxim IC, San Jose, CA, USA), microcontroller (C8051F320; Silicon Labs, Austin, TX, USA), and a heating device, which can adjust automatically and allow real-time monitoring and recording of water temperature [38], so the diurnal variation in the water temperature in the heated mesocosm varied in the same way as the control, and was always 3.5 °C higher than the control. The temperature sensors were located 0.5 m below the water surface in both the unheated and heated mesocosm, while the heating devices (600 W; A10-3, Xin Shao Guang, Wuhan, China) were installed 30 cm below the water surface in each heated mesocosm. For eutrophication treatments, nitrogen (N) and phosphorus (P) were introduced in a mass ratio of 10:1, achieved by dissolving NaNO_3_ and KH_2_PO_4_ powder in de-mineralized water, respectively. The nutrient loading doses for nitrogen ranged from 0.25 to 1.6 mg L^−1^, and for phosphorus, they ranged from 0.025 to 0.16 mg L^−1^. Pesticide treatment involved adding imidacloprid with 70% active ingredients, with an average dose of 32.67 µg L^−1^ and a range of 10–50 µg L^−1^. Nutrient and pesticide additions were conducted biweekly, with doses adjusted based on agricultural activities and precipitation intensity in the region, to simulate a more realistic scenario with temporally changing multiple stressors (Figure 1).

### 2.2. Sample Collection and Processing

The experimental system functioned from 13 March 2021 to 7 November 2021, with sample collection beginning on 22 April 2021. Conductivity and pH were measured bi-weekly using a HACH HQD portable meter (HQ40d, HACH, Ames, IA, USA), and turbidity was measured bi-weekly using a portable WGZ-2B turbidimeter (Shanghai, China). At the end of the experiment, water samples were collected with a tubular water sampler (70 mm in diameter and 1 m in length) for the analysis of total nitrogen (TN), total phosphorus (TP) concentrations, and phytoplankton biomass (expressed as chlorophyll A (Chl-a) concentrations). The water samples with 100 mL for TN and TP analysis were digested with potassium persulfate and analyzed using spectrophotometry. The water samples with 100–500 mL (depending on the solid matter content in the water column) for Chl-a analysis were filtered through 47 mm Whatman filter papers (GF/C filter, Whatman, Kent, UK) and determined using spectrophotometric analysis after acetone extraction [39]. There were three replicates for all analyses in each mesocosm.

Water samples with 100 mL for protozoan analysis were collected monthly. After collection, the samples were immediately fixed in Lugol’s iodine solution for subsequent identification and counting. After the samples were deposited for more than 48 h, they were concentrated to 10 mL, and then finally 1 mL samples were taken to the zooplankton counting chamber for the identification and quantification of protozoa under an upright microscope (OLYMPUS CX-31, 100–100×, OLYMPUS, Tokyo, Japan). Each sample was counted twice. Species were identified with reference to Shen [26] and Foissner et al. [40]. Five functional groups (A, B, A&B, R, N) were defined based on the body size, predation pattern, and prey of protozoa [25]. The functional groups whose abundance or biomass accounted for more than 10% of the treatment were regarded as the dominant functional groups [26]. The biomass of the protozoa was converted using the volume method because of their small individual sizes [41].

### 2.3. Data Processing and Analysis

The Shannon–Wiener diversity index (*H’*), Pielou’s evenness index (*J*), and Simpson index (*D*) were used to describe the community structure of the protozoa [42], which were calculated by the Vegan package in R software R_4.2.3 (R Core Team, 2022). The formulas for calculating each index are as follows:

*H’* = −∑*N i* = 1 *Ni*·ln*Ni*;*J* = *H*’/ln(*S*);*D* = ∑*n*(*n* − 1)/*N*(*N* − 1).


Here, *n* is the number of individuals in the sample, *N* is the total number of individuals in the sample, *S* is the number of species in the community, and *Ni* is the proportion of the sample represented by species *i*.

Referring to Zhang et al. [43], a generalized linear model (*lmer* function from the *lme4* package in software R_4.2.3) was used to analyze the impacts of different treatments (temperature rise, eutrophication and insecticides) on the chlorophyll-A concentration and community structure of protozoa (abundance and biomass of protozoa, diversity index of protozoa, and relative abundance of different functional groups).

One-way ANOVA (analysis of variance) was used to compare the effects of environmental factors of the different treatments on the community structure of protozoa (including abundance and biomass of protozoa, diversity index of protozoa, and relative abundance of different functional groups) by using the *car* package in R_4.2.3. Data that did not conform to the normal distribution were using lg (*x* + 1).

For the above analysis, the “tukey test” function from *emmeans* package in R software R_4.2.3 was used to evaluate the statistical significance, and the *ggplot2* package in software R_4.2.3 and Origin 8.5 were used for drawing graphics.

## 3. Results

### 3.1. Effects of Different Treatments on Physical and Chemical Indexes of the Water Body

During the experiment, the average water temperature of the environmental control group C was 26.5 ± 4.1 °C (mean ± standard error), whereas the average water temperature of the continuous warming group W was 30.1 ± 4.0 °C, and the average temperature of group W was 3.5 °C higher than that of control group C during the experiment (Figure 2).

Water quality parameters in the different treatments at the end of the experiment are shown in Table 1. The concentration ranges of TN and TP were 1.66–3.98 mg/L and 0.03–0.36 mg/L, respectively. For TN and TP, the treatment with the highest concentration was EP. The range of dissolved oxygen was 8.11–9.87 mg/L, and the warming treatments were generally lower than the non-warming treatments. The ranges of pH and conductivity across all treatments were 7.56–8.96 and 159.35–261.00 μS/cm, respectively. The concentration of Chl-a in the different treatment groups ranged from 15.56 to 249.30 μg/L, among which the EP treatment group had the highest concentration of Chl-a (249.30 μg/L), followed by the WEP group (198.00 μg/L), and then the W treatment group had the lowest concentration (15.56 μg/L). The concentrations of Chl-a in the different treatment groups were significantly different (*p* < 0.05), with the E and P groups being significantly higher than the C and W groups, and the EP and WEP groups being significantly higher than the E and P groups. Nitrogen and phosphorus addition, pesticide addition, and continuous warming had an interactive effect on the concentration of Chl-a. Under the three conditions of adding nitrogen and phosphorus, adding pesticides, and simultaneously adding nitrogen and phosphorus with pesticides, the concentration of Chl-a was lower in the continuous warming groups (WE, WP, WEP) compared to the no-warming groups (E, P, EP) (Figure 3).

### 3.2. Effects of Different Treatments on Community Structure of Protozoa

A total of 46 species of protozoa from 36 genera were identified in the simulation systems, including 26 species of ciliates from 22 genera and 20 species of amoeba from 14 genera (Table 2). Most amoebae were algivores, bacterivores, or both. In addition to the above three functional groups, some ciliates were predators (Table 2).

There were differences in the abundance and biomass of protozoa among different treatment groups, and each treatment group was higher than the control group C. Among them, the WEP treatment group had the highest abundance and biomass of protozoa, at 5.42 × 10^4^ ind./L and 4.78 mg/L, respectively and the control group C had the lowest, at 1.13 × 10^4^ ind./L and 1.06 mg/L, respectively (Figure 4). Multivariate analysis of variance showed that the biomass of protozoa was significantly increased in the eutrophication (E), pesticide (P), warming and eutrophication (WE) along with warming and pesticide (WP) treatment groups (*p* < 0.05) (Table 3).

The addition of eutrophication (E) and pesticide (P) significantly increased protozoan abundance. The effect of P was more significant (*p* < 0.01) (Table 3); the abundance of protozoa in treatment E was 456.25 ind./L higher than that in control group C and that of treatment P was 2383.33 ind./L higher than that of control group C (Figure 4). Among the added-pesticide groups (P, EP, WP, and WEP), the WP group had the lowest abundance of protozoa (Figure 4). From the perspective of biomass, all treatments had a certain effect on improving the biomass of protozoa, with treatment groups P, WE, WP, and WEP having significant effects, among which the WEP group had the highest biomass, followed by the P and EP groups (Figure 4).

Differences in the community structure of protozoa in the different treatments were characterized using the Simpson diversity, Shannon–Wiener diversity, richness, and Pielou’s evenness indices. The four indices showed certain differences in the different treatments (Figure 5): the Simpson diversity, Shannon–Wiener diversity, and richness indices of the eutrophication and pesticide treatment (EP) were the highest, with values of 0.64 ± 0.16, 1.66 ± 0.48, and 4.75 ± 1.75, respectively, while the pesticide treatment (P) had the lowest diversity indices of 0.47 ± 0.14, 1.23 ± 0.32, 3.00 ± 1.00, and 0.73 ± 0.11, respectively (Figure 5). The significance analysis showed that eutrophication (E) had a significant impact on all four diversity indices (*p* < 0.05), among which the richness index was significantly increased (*p* < 0.001). The warming and pesticide treatment (WP) had a significant negative impact on richness index, Shannon–Wiener diversity index, and Pielou’s evenness index (all *p* < 0.05). In addition, both the pesticide treatment (P) and the eutrophication and pesticide treatment (EP) also had a significant negative impact on Pielou’s evenness index (all *p* < 0.05) (Table 4).

### 3.3. Effects of Different Treatments on Functional Group Composition of Protozoa

A total of nine species of dominant functional protozoa were observed during the eight-month experiment including two algivores of *Strobilidium* sp. (Figure 6A) and *Difflugia* sp. (Figure 6B), two bacterivores of *Cyclidium* sp. (Figure 6C) and *Halteria* sp. (Figure 6D), three both algivores and bacterivores of *Cyrtolophosis* sp. (Figure 6E), *Tintinnidium* sp. (Figure 6F) and *Tintinopsis* sp. (Figure 6G), and two predators of *Didinium* sp. (Figure 6H) and *Podophrya* sp. (Figure 6I). Among them, *Strobilidium* sp. and *Halteria* sp. occurred almost throughout the year, while *Difflugia* sp., *Tintinnidium* sp., *Tintinopsis* sp., and *Podophrya* sp. only occurred in certain months (Table 5). Overall, dominant species differ according to treatment conditions, take the data at the end of the experiment as an example (Table 6), *Strobilidium* sp. was distributed in almost all treated or untreated systems with very high relative abundances (36–76%), *Didinium* sp. only occur in the “P” and “WP” treatment system but with high relative abundances (54% and 37%, respectively), and *Halteria* sp., while *Tintinopsis* sp. only occur in the control system with low relative abundance (7%).

From the perspective of abundance, the main dominant functional group that appeared in most treatment groups was algivores, the relative abundance of which exceeded 50%. However, in both pesticide treatment (P) and the warming and pesticide treatment (WP), the main dominant functional group was predators, the relative abundance of which also exceeded 50% (Figure 7). Due to the relative abundance ratio of nonselective omnivores (N) being as low as 0.36%, the N group was not included in the functional group analysis.

It can be seen from the trend of the variation in the relative abundance of the dominant groups that the abundance of algivores increased significantly in the W treatment group (*p* < 0.05), while it decreased in the P (*p* < 0.01) and WP (*p* < 0.001) treatment groups (Figure 7 and Table 7). Meanwhile, the abundance of bacterivores was significantly increased in the WP (*p* < 0.05) and WEP (*p* < 0.01) treatment groups; the abundance of species that were both algivores and bacterivores were significantly decreased in the P, EP and WP treatment groups (all *p* < 0.05); the abundance of predators was significantly increased in the P (*p* < 0.01), WP (*p* < 0.01) and WEP (*p* < 0.05) treatment groups (Figure 7 and Table 7).

## 4. Discussion

### 4.1. Effects of Warming, Eutrophication, and Pesticide Pollution on the Community Structure of Protozoa

Climate change and other co-occurring large-scale environmental changes (such as water eutrophication) can cause remarkable changes in the community structure of protozoa in lake ecosystems [44,45,46,47]. The results of this study also indicate that continuous warming, eutrophication, and pesticides have certain effects on the community structure of protozoa and that these three environmental factors can interact and produce a comprehensive impact on the protozoan community (Figure 8).

In this study, we found that warming (W) could increase the abundance and biomass of protozoa, but the effect was not significant (Figure 4; Table 3), and there was no obvious effect on α diversity. To some extent, warming may increase the metabolic rate of protozoa, resulting in an increased growth rate, abundance, and biomass [48,49]; however, owing to the wide adaptability of protozoa to temperature [50], the single warming treatment in this study did not significantly affect the protozoan community structure and diversity. Meanwhile, warming combined with pesticides (WP) or eutrophication (WE) did significantly affect the biomass and diversity of protozoa (Table 3 and Table 4)—see below for details.

Eutrophication (E) had significant effects on protozoan abundance, biomass, and diversity, which significantly increased the abundance, biomass and α diversity (including the Simpson, Shannon, and richness indices) of protozoa (Figure 4 and Figure 5; Table 3 and Table 4). Under eutrophication, abundant nitrogen and phosphorus nutrients can cause a large number of phytoplankton and bacteria to proliferate. Protozoa, as the next trophic level of phytoplankton and bacteria, can develop rapidly when plentiful food resources are supplied, so that algivores and bacterivores can easily gain advantages in species competition [51], and the community structure can change from simple to complex [52,53,54]. Global change and eutrophication have an interactive effect on zooplankton [44]. Our results also showed that the combined effect of eutrophication and warming (WE) led to a significant change in the protozoan community structure. Compared with the eutrophication treatment (E), the combined effect of the WE treatment only significantly increased the biomass, but not the abundance of protozoa, which may be caused by the difference in the individual sizes of dominant taxa in different treatments [55]. Simultaneously, the combined effects of the WE treatment on α diversity (Shannon and richness indices) were weakened, which might be due to the interaction between abundant nutrition (nitrogen and phosphorus) and appropriate temperature. Although it could further increase the total amount of protozoa [46,56], the dominant species developed faster but limited the improvement of species richness in the whole community.

This study showed that the pesticide treatment (P) increased the abundance and biomass of protozoa while reducing the α diversity (including Simpson, Shannon and Pielou indices) and changing the community structure from complex to simple (Table 3 and Table 4; Figure 4 and Figure 5). Many studies have found that pesticides may have toxic effects on aquatic organisms such as protozoa [57]. By affecting antioxidant enzymes and pathological tissue formation, they can change the growth and reproduction rules of aquatic organisms, and then affect their community structure and biodiversity [58,59,60]. There are differences in the concentration and toxicity of various pesticides on different groups of aquatic organisms [61]. Studies have shown that pesticides (imidacloprid) at 12 μg/L can remarkably reduce the hatching and growth rate of soil amoeba spores (*Dictyostelium discoideum*) [62], leading to a decrease in the total amount of protozoa. We found that the total number of protozoa did not decrease, but the specific community composition changed greatly, with a large number of ciliates and few amoebae observed in pesticide treatment. The reason for this phenomenon may be that ciliates and amoeba have different sensitivity to pesticides, with amoeba being more susceptible to pesticides [61,62], and the pesticide concentrations designed in this experiment were insufficient to produce negative effects on the growth of ciliates. When the combination of pesticides and warming (WP) was continuously applied, the α diversity of protozoa decreased, the biomass was relatively low, and the overall community structure was simple (Figure 4 and Figure 5). Consequently, there was a synergistic effect between the combined treatment of pesticides and warming (WP), and the toxicity of pesticides (imidacloprid) increased with increasing temperature [63,64,65], resulting in a decrease in the total amount of protozoa compared with the pesticides treatment alone (P) (Figure 4). In addition, although the abundance and biomass of protozoa increased in the eutrophication, pesticides and warming treatment (WEP), the results of correlation analysis showed that the effect of the WEP treatment on the protozoan community structure was not significant, indicating that the combined effect of these three environmental factors slowed down the effect of each single environmental factor. The specific mechanism needs to be further explored.

### 4.2. Effects of Warming, Eutrophication, and Pesticide Pollution on Protozoan Functional Groups

Our results showed that there was no significant difference in protozoan functional group composition among the treatment groups (Figure 7), but the dominant functional group was significantly affected by different environmental factors (Table 7). Under the combined treatments of warming and other environmental changes (WP and WEP), bacterivores were the dominant functional group, and their relative abundance in the WEP treatment increased by 23.39% compared with the control group (Figure 7). The abundance of bacteria in aquatic ecosystems was significantly positively correlated with water temperature [66], and the increase in water temperature promoted the growth of bacteria [67]. This may be due to bacterivores, which exclusively feed on bacteria, taking advantage of the abundant food resources to replace some of the other functional groups due to the large number of bacteria in the warming conditions.

Our results also showed that eutrophication (E) had no significant effect on protozoa functional group composition (Table 7). However, when eutrophication was combined with other environmental changes (EP and WEP), the Chl-a concentration increased sharply (Figure 3), algivores became the largest dominant group, the proportion of predators increased, and the proportion of species that were both algivores and bacterivores decreased (Figure 7). The high Chl-a concentration in the EP and WEP group may be because, under eutrophication, the addition of N and P would proliferate the growth of phytoplankton, and also because protozoa are more sensitive to imidacloprid than phytoplankton, the removal of protozoa especially algivorous protozoa would contribute for algal proliferation. Under the EP and WEP treatments, algivores became the dominant functional group, which was mainly due to the large increase in phytoplankton promoted by the two treatment groups (Figure 3; Chl-a concentration was the highest and second highest, at 249.30 μg/L and 198.00 μg/L, respectively). Thus, algivores with efficient feeding ability of phytoplankton gained growth advantages [27]. Simultaneously, predators promoted their development by feeding on small algivores [68].

Under pesticide treatment (P) and its combination with warming (WP), the proportions of algivores and species that were both algivores and bacterivores decreased, whereas predators significantly increased and became the dominant group. Compared with the control treatment (C), the relative abundance of predators in the P and WP treatments increased by 50.77% and 49.56%, respectively (Figure 7). It has been theorized that pesticides can reduce the biological activity of protozoa, or even poison them and phytoplankton, leading to death [61,62,69,70]. The results of our study showed that algivores and species that were both algivores and bacterivores decreased, and predators increased significantly, under the treatment of pesticides; the reasons may be similar to those mentioned above. Firstly, after pesticides enter the aquatic system, sensitive protozoa and some phytoplankton may be directly poisoned [71], resulting in the reduction in algivores and species that were both algivores and bacterivores that feed on phytoplankton. Secondly, pesticides can promote the proliferation of toxic cyanobacteria [70], which have a toxic effect on algivores and species that were both algivores and bacterivores. This can result in the reduction in both these groups through the bottom-up effect, thus affecting the protozoan functional structure. Thirdly, most predators in the experimental system were large ciliates, such as *Didinium balbianii nanum* and *Didinium nasutum* (Table 2), which are more tolerant to pesticides [72]; they are more likely to feed on other protozoa such as *Paramecium* to gain advantages [73]. It can be seen that changes in the composition of the functional groups of protozoa and their dominant taxa are not only directly affected by environmental factors such as water temperature, eutrophication state, and toxic pollution, but also indirectly affected by the fluctuations in other aquatic taxa such as phytoplankton. This phenomenon clarifies the relationship between protozoa and other environmental factors in water in detail. It also reflects the changes in the ecological functions of protozoa under different circumstances.

### 4.3. Study Limitations and Directions for Future Exploration

Our experiments showed that the combined effects of different environmental stressors form interactions in shallow lake ecosystems, thus making the changes in the aquatic environment and protozoan community more complex. In future work, additional monitoring studies with concentration gradients and longer sequences should be run. This will allow workers to explore the thresholds of warming, nitrogen and phosphorus addition, and pesticide pollution on protozoa. It will also inform us as to the responses of the protozoan community structure and functional groups to multiple environmental stressors. Such studies would provide a more comprehensive scientific basis for the protection and restoration of shallow lake ecosystems. However, additional taxa should be included in future studies; these would include cladocerans, copepods, and rotifers.

## 5. Conclusions

Based on mesocosm experiments, this study simulated shallow lake ecosystems under toxic and harmful pollution stresses such as global warming, eutrophication, and pesticides, and evaluated the combined effects of multiple environmental stressors on protozoan community structure and functional groups. The conclusions can be summarized as follows: (1) Both eutrophication and pesticides had a considerable promotional effect on the abundance and biomass of protozoa, while the effect of warming was not considerable, but when warming and eutrophication were combined, there was a synergistic effect and the biomass of protozoa was significantly increased; when warming and pesticides were combined, there was an antagonistic effect between the two treatments, resulting in a decrease in the abundance and biomass growth of protozoa; (2) Eutrophication greatly promoted the α diversity of protozoa, and affected the species richness and dominant species composition of the protozoan community; the combined treatment of warming and pesticides remarkably reduced the α diversity of protozoa; (3) Warming, eutrophication, and pesticides were important factors affecting the functional groups of protozoa, with the effects of warming and pesticides being more significant.

This study only focused on a single level of eutrophication and pesticide pollution. To provide a more scientific basis for the protection and restoration of aquatic ecosystems in shallow lakes, it is recommended to incorporate various levels of eutrophication and pesticides and explore thresholds and mechanisms of the effects on protozoa. Furthermore, it is advised to consider the temporal dynamics of the complex effects of multiple environmental stressors in future studies. Observations at multiple time points will provide a more holistic assessment of the integrated impacts of multiple stressors over extended time scales.

## Figures and Tables

**Figure 1 animals-14-01293-f001:**
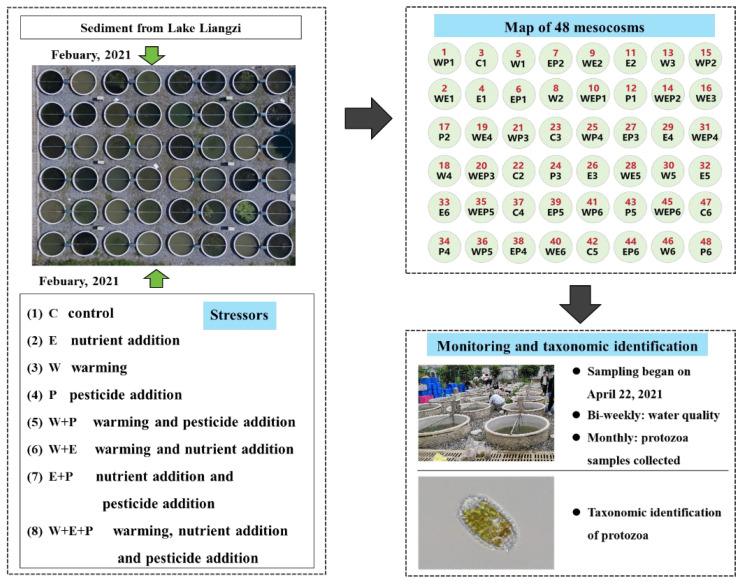
Schematic of the experimental design, monitoring, and taxonomic identification of the mesocosms.

**Figure 2 animals-14-01293-f002:**
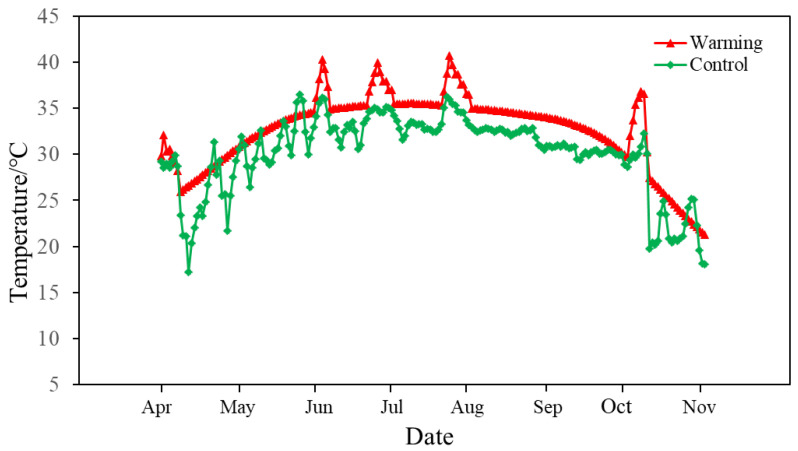
Seasonal changes in average daily water temperature for the control and warming treatments during the experiment.

**Figure 3 animals-14-01293-f003:**
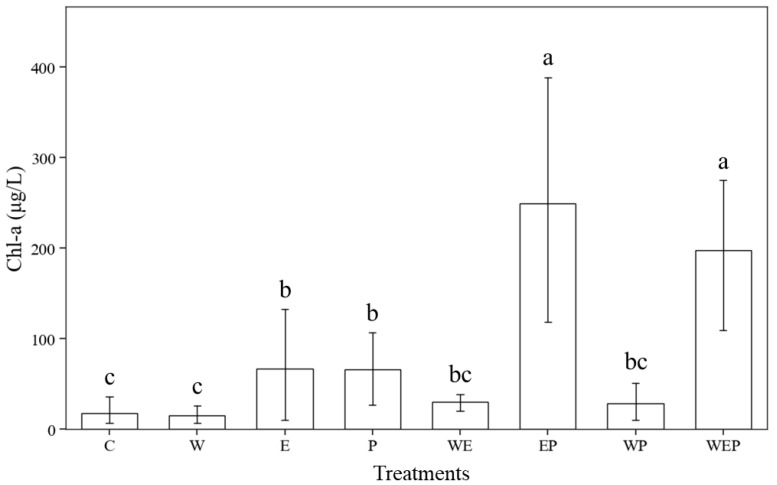
Contents of chlorophyll-A (chl-a) in different treatments at the end of the experiment. C: control; W: warming; E: eutrophication; P: pesticide; WE: warming and eutrophication; EP: eutrophication and pesticide; WP: warming and pesticide; WEP: warming, eutrophication and pesticide. Lowercase letters represent significant differences in means between different treatments (post hoc tests, *p* < 0.05), while the same letters indicate no significant differences. Vertical bars are standard errors.

**Figure 4 animals-14-01293-f004:**
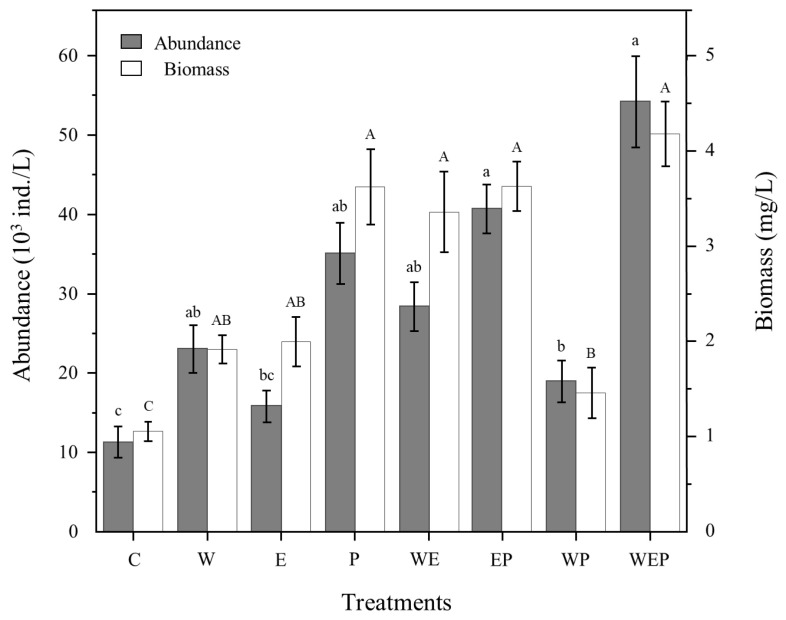
Abundance and biomass of protozoa of different treatments at the end of experiment. C: control; W: warming; E: eutrophication; P: pesticide; WE: warming and eutrophication; EP: eutrophication and pesticide; WP: warming and pesticide; WEP: warming, eutrophication and pesticide. Capital letters represent significant differences in means between abundance of different treatments (post hoc tests, *p* < 0.05), lowercase letters represent significant differences in means between biomass of different treatments (post hoc tests, *p* < 0.05) while the same letters indicate no significant differences. Vertical bars are standard errors.

**Figure 5 animals-14-01293-f005:**
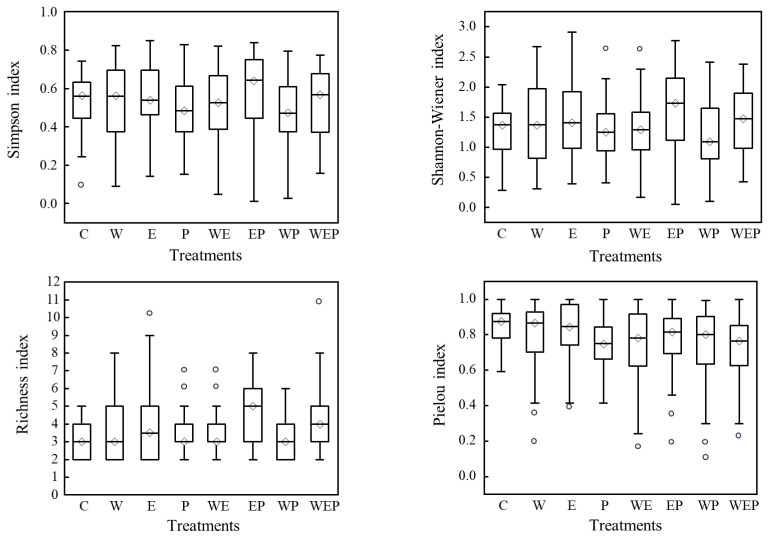
Comparison of protozoa diversity indices in different treatments at the end of experiment. C: control; W: warming; E: eutrophication; P: pesticide; WE: warming and eutrophication; EP: eutrophication and pesticide; WP: warming and pesticide; WEP: warming, eutrophication and pesticide. Note: The rectangular columns in the figure indicate the ranges of the measured indices; the circles in the figure indicate the outliers of the measured indices; the diamonds located at the line in the rectangular columns indicate the means of the measured indices).

**Figure 6 animals-14-01293-f006:**
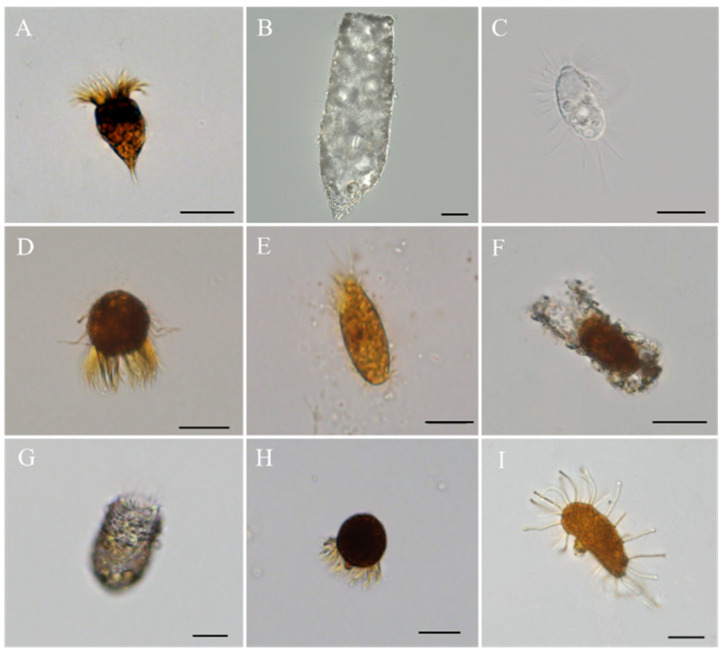
Dominant functional groups of protozoa in mesocosms. (**A**), *Strobilidium* sp.; (**B**), *Difflugia* sp; (**C**), *Cyclidium* sp.; (**D**), *Halteria* sp.; (**E**), *Cyrtolophosis* sp.; (**F**), *Tintinnidium* sp.; (**G**), *Tintinopsis* sp.; (**H**), *Didinium* sp.; (**I**), *Podophrya* sp.; Scale, 20 μm.

**Figure 7 animals-14-01293-f007:**
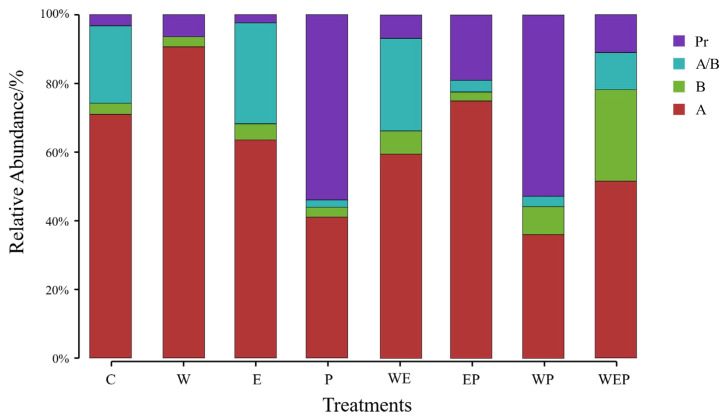
Relative abundance of protozoan functional groups in different treatments at the end of experiment. C: control; W: warming; E: eutrophication; P: pesticide; WE: warming and eutrophication; EP: eutrophication and pesticide; WP: warming and pesticide; WEP: warming, eutrophication and pesticide. A: algivores; B: bacterivores; A&B: algivores and bacterivores; Pr: predators.

**Figure 8 animals-14-01293-f008:**
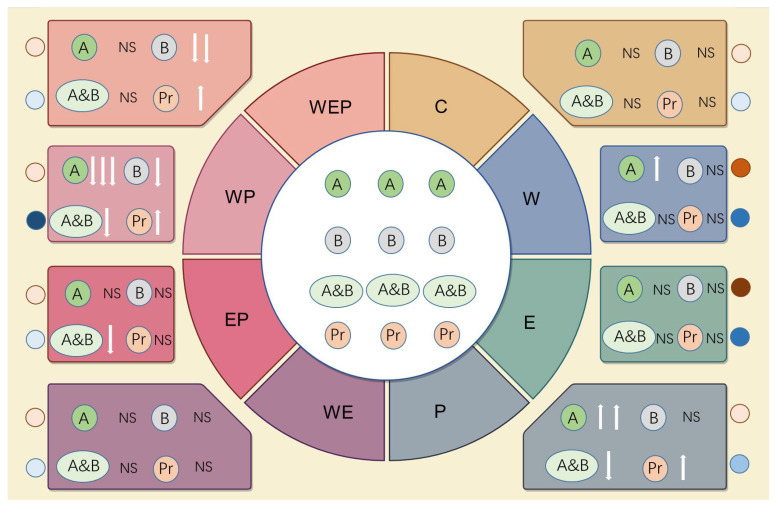
A systematic flowchart about the responses of protozoa on the effects of warming, eutrophication and pesticides. A: algivores; B: bacterivores; A&B: algivores and bacterivores; Pr: predators. Arrows show the responses of functional groups; up arrows show the positive responses; down arrows show the negative responses. NS: not significant (*p* > 0.1); one arrow indicating moderate significance (0.01 < *p* ≤ 0.05); two arrows indicating significance (0.001 < *p* ≤ 0.01); three arrows indicating high significance (*p* ≤ 0.001). Circles show the response of communities; circles in brown show the changes in abundance; circles in blue show the changes in biomass. The circles in light brown or blue showed no significance (*p* > 0.1); the circles in normal brown or blue showed a moderate significant increase (0.01 < *p* ≤ 0.05); the circles in dark brown or blue showed a significant increase (0.001 < *p* ≤ 0.01); the circles in the darkest brown or blue showing a high significant increase (*p* ≤ 0.001).

**Table 1 animals-14-01293-t001:** Water quality parameters in different treatments at the end of the experiment (mean ± SE).

Treatments	TN (mg/L)	TP (mg/L)	DO (mg/L)	pH	Cond. (μS/cm)
C	1.66 ± 0.58	0.03 ± 0.02	8.96 ± 0.51	7.56 ± 0.18	179.35 ± 18.86
W	1.98 ± 0.41	0.03 ± 0.02	8.51 ± 0.39	7.85 ± 0.51	214.48 ± 11.73
E	1.90 ± 0.70	0.09 ± 0.10	9.18 ± 0.49	8.11 ± 0.46	206.95 ± 20.94
P	1.71 ± 0.44	0.05 ± 0.04	9.87 ± 0.40	8.61 ± 0.67	159.35 ± 8.65
WE	2.47 ± 0.53	0.14 ± 0.14	8.11 ± 0.20	7.97 ± 0.56	254.17 ± 19.58
EP	3.98 ± 1.60	0.36 ± 0.20	9.38 ± 1.11	8.81 ± 0.65	206.95 ± 21.73
WP	1.78 ± 0.29	0.06 ± 0.04	8.54 ± 0.49	8.29 ± 0.45	187.47 ± 11.11
WEP	3.90 ± 0.87	0.35 ± 0.19	8.25 ± 2.55	8.96 ± 0.59	261.00 ± 13.29

C: control; W: warming; E: eutrophication; P: pesticide; WE: warming and eutrophication; EP: eutrophication and pesticide; WP: warming and pesticide; WEP: warming, eutrophication, and pesticide; TN: total nitrogen; TP: total phosphorus; DO: dissolved oxygen; Cond.: conductivity.

**Table 2 animals-14-01293-t002:** Species of protozoa and functional groups occurring in mesocosms.

Scientific Name	Functional Group	Scientific Name	Functional Group
*Vorticella* sp.	B	*Raphidiophrys* sp.	A&B
*Cyclidium* sp.	B	*Acanthocystis* sp.	A&B
*Strobilidium gyrans*	A	*Podophrya* sp.	Pr
*Strobilidium* sp.	A	*Prorodon* sp.	A
*Halteria* sp.	B	Order Hymenostomata	N
*Askenasia* sp.	B	*Holophrya* sp.	B
*Mesodinium* sp.	B	*Coleps* sp.	A&B
*Tintinnidium* sp.	A&B	Order Prostomatida	N
*Tintinnidium pusillum*	A&B	*Lagynaphrya conifera*	A
*Pseudodifflugia* sp.	A	*Cyrtolophosis* sp.	A&B
*Difflugia acuminata*	A	*Pseedoglaucoma* sp.	B
*Difflugia* sp.	A	*Lacrymaria* sp.	Pr
*Centropyxis aculeata*	A&B	*Dileptus* sp.	B
*Centropyxis hemisphaerica*	B	*Nassula* sp.	A
*Tintinopsis* sp.	A&B	*Trachelophyllum* sp.	B
*Tintinnopsis niei*	A&B	Order Hypotrichida	N
*Tintinnopsis wangi*	A&B	*Didinium* sp.	Pr
*Leprotintinnus* sp.	A&B	*Didinium balbianii nanum*	Pr
*Cyclopyxis eurostoma*	A	*Didinium nasutum*	Pr
*Arcella hemisphaerica*	A&B	*Litonotus* sp.	Pr
*Pontigulasia incisa*	A	*Urotricha* sp.	A
*Enchelys* sp.	A&B	*Trochilia palustris*	A&B
Family Amoebidae	N	*Strombidium* sp.	Pr
Family Vahlkampfiidae	N	*Actinobolina* sp.	Pr
*Nuclearia* sp.	A&B	*Aspidicca costatas*	B

A: algivores; B: bacterivores; A&B: algivores and bacterivores; Pr: predators; N: nonselective omnivores.

**Table 3 animals-14-01293-t003:** Effects of warming, eutrophication, pesticide, and their interactions on protozoan abundance and biomass.

Treatments	Abundance			Biomass		
	*F*	*Df*	*p*	*F*	*Df*	*p*
W	1.54	1	0.2198	1.01	1	0.3183
E	5.78	1	**0.0195 ***	9.85	1	**0.0027 ****
P	8.67	1	**0.0047 ****	7.26	1	**0.0093 ****
WE	0.03	1	0.8664	4.22	1	**0.0447 ***
EP	1.24	1	0.2695	0.01	1	0.9093
WP	2.81	1	0.0994	12.08	1	**0.001 *****
WEP	0.46	1	0.4984	1.95	1	0.1682

Bold numbers indicate *p* < 0.05 (post hoc tests); “*”, “**” and “***” represent *p* less than 0.05, 0.01 and 0.001 (post hoc tests), respectively. W: warming; E: eutrophication; P: pesticide; WE: warming and eutrophication; EP: eutrophication and pesticide; WP: warming and pesticide; WEP: warming, eutrophication, and pesticide. Note: Data for this analysis was from the last sampling at the end of the experiment.

**Table 4 animals-14-01293-t004:** Effects of warming, eutrophication, pesticide, and their interactions on protozoan diversity indices ^#^.

Treatments	Simpson Index			Shannon-Wiener Index			Richness Index			Pielou Index		
	*F*	*Df*	*p*	*F*	*Df*	*p*	*F*	*Df*	*p*	*F*	*Df*	*p*
W	0.11	1	0.7380	0.69	1	0.4088	2.47	1	0.1214	1.54	1	0.2194
E	9.37	1	**0.0034 ****	11.24	1	**0.0014 ****	28.18	1	**<0.001 *****	5.20	1	**0.0264 ***
P	1.00	1	0.3227	0.07	1	0.7925	0.79	1	0.3783	9.07	1	**0.0039 ****
WE	0.03	1	0.8538	0.01	1	0.9161	0.39	1	0.5358	0.18	1	0.6711
EP	0.52	1	0.4749	0.01	1	0.9192	0.04	1	0.8431	4.12	1	**0.0471 ***
WP	3.98	1	0.0509	5.53	1	**0.0222 ***	6.19	1	**0.0158 ***	6.52	1	**0.0134 ***
WEP	0.04	1	0.8338	0.05	1	0.8291	0.43	1	0.5147	0.14	1	0.7104

^#^ The data for this analysis were from the whole year including eight samples, and using the mean from six replicates for each control or treatment. Bold numbers indicate *p* < 0.05 (post hoc tests); “*”, “**” and “***” represent *p* less than 0.05, 0.01 and 0.001 (post hoc tests), respectively. W: warming; E: eutrophication; P: pesticide; WE: warming and eutrophication; EP: eutrophication and pesticide; WP: warming and pesticide; WEP: warming, eutrophication and pesticide.

**Table 5 animals-14-01293-t005:** Dominant functional groups of different months in mesocosms.

Functional Group	A	A	B	B	A&B	A&B	A&B	Pr	Pr
Dominant Species	*Strobilidium* sp.	*Difflugia* sp.	*Cyclidium* sp.	*Halteria* sp.	*Cyrtolophosis* sp.	*Tintinnidium* sp.	*Tintinopsis* sp.	*Didinium* sp.	*Podophrya* sp.
April	+			+		+			
May	+			+					
June	+	+		+					
July	+			+	+				
August	+			+	+				
September	+		+	+	+			+	
October	+			+	+			+	
November	+		+			+	+		+

A: algivores; B: bacterivores; A&B: algivores and bacterivores; Pr: predators. “+” indicating the corresponding species was dominant species.

**Table 6 animals-14-01293-t006:** Relative abundance of dominant protozoan functional species in different treatments at the end of experiment.

Functional Group	Scientific Name	Treatment							
		C	W	E	P	WE	EP	WP	WEP
A	*Strobilidium* sp.	0.3826	0.7636	0.6667	0.4103	0.5951	0.7500	0.3605	0.5152
B	*Cyclidium* sp.		0.1818					0.0815	0.2662
A&B	*Tintinopsis* sp	0.0696							
A&B	*Tintinnidium* sp.	0.0522		0.2099		0.2699			0.1084
R	*Didinium* sp.				0.5400			0.3691	
R	*Podophrya* sp.						0.1034	0.1545	

**Table 7 animals-14-01293-t007:** Effects of treatment on functional groups of protozoa relative abundance ^#^.

Functional Group	A			B			A&B			Pr		
Treatments	*F*	*Df*	*p*	*F*	*Df*	*p*	*F*	*Df*	*p*	*F*	*Df*	*p*
W	6.85	1	**0.0151 ***	0.91	1	0.3494	0.56	1	0.4583	0.32	1	0.7463
E	3.69	1	0.0664	3.85	1	0.0612	0.06	1	0.8386	0.26	1	0.7968
P	7.88	1	**0.0097 ****	0.05	1	0.9369	6.86	1	**0.0164 ***	8.88	1	**0.0081 ****
WE	2.08	1	0.1614	4.20	1	0.0513	0.79	1	0.3816	0.60	1	0.5489
EP	0.62	1	0.4386	0.61	1	0.4424	7.02	1	**0.0158 ***	0.60	1	0.5532
WP	22.16	1	**<0.001 *****	4.78	1	**0.0386 ***	2.38	1	**0.0255 ***	10.30	1	**0.0029 ****
WEP	0.21	1	0.6482	13.15	1	**0.0013 ****	2.28	1	0.1436	6.77	1	**0.0105 ***

^#^ The data for this analysis were from the end of the experiment. The original data were the relative abundances of each functional group in different treatments, and six replicates were regarded as individual. Bold numbers indicate *p* < 0.05 (post hoc tests); “*”, “**” and “***” represent *p* less than 0.05, 0.01 and 0.001 (post hoc tests), respectively. W: warming; E: eutrophication; P: pesticide; WE: warming and eutrophication; EP: eutrophication and pesticide; WP: warming and pesticide; WEP: warming, eutrophication, and pesticide. A: algivores; B: bacterivores; A&B: algivores and bacterivores; Pr: predators.

## Data Availability

All data are contained within the article.

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
