# Peer review of "Responses of Protozoan Communities to Multiple Environmental Stresses (Warming, Eutrophication, and Pesticide Pollution)"

_animals, 2024, doi:10.3390/ani14091293_

Round 1
Reviewer 1 Report
Comments and Suggestions for Authors
The present study investigated the effects of warming, eutrophication and pesticide pollution on protozoa. The results are sound and the discussion is comprehensive. I think it will be interesting for readers no only specialist of Protozoa but also environment researchers. I only have some minor comments that authors should address:
Line 60, how can use of fossil fuels affect aquatic ecosystem?
Is there any research on eutrophication and pesticide pollution affecting aquatic communities? If yes, authors need to provide in the introduction.
Authors should make the second paragraph of the introduction briefly and concise. The present version is too redundant.
Line 114, 10cm deep?
Did authors pretreat the sediments before fill them to the mesocosm?
In the text, author wrote the mesocosm was filled with tap water, but in the figure 1, it is water from Lake Nanhu.
Authors collected 100ml for protozoa analysis and then concentrated before counting and identification. But they did not provide the proportion of the samples were examined. And if there were replicates please provide the information as well.
Why did EP group have the highest concentration of Chl-a? Pestcide should be harmful to algae too. Authors need to explain it in the discussion.
Line 352, I cannot understand “the protection of sufficient food resources”, please reword it.
It is weird that in Figure 3 all error bars looks the same, Please check your data.
Provide error bars for figure 4
Author Response
The present study investigated the effects of warming, eutrophication and pesticide pollution on protozoa. The results are sound and the discussion is comprehensive. I think it will be interesting for readers no only specialist of Protozoa but also environment researchers. I only have some minor comments that authors should address:
Comment 1: Line 60, how can use of fossil fuels affect aquatic ecosystem?
Response 1: We have removed the phrase of “fossil fuels” and the new sentence was as follows “Simultaneously, owing to the production and use of chemical fertilizers, as well as the discharge of rural and urban sewage, lake eutrophication has become a serious cause for concern.” (Lines 50-53).
Comment 2: Is there any research on eutrophication and pesticide pollution affecting aquatic communities? If yes, authors need to provide in the introduction.
Response 2: We have added relevant introductions and references, as “Previous researches have shown that climate warming, nutrients, and pesticide pollution had significant individual and interactive effects on aquatic ecosystems and its food web (Chará-Serna et al., 2019; Wang et al., 2023)”. (Lines 57-60)
Chará-Serna, A.M., Epele, L.B., Morrissey, C.A., Richardson, J.S. Nutrients and sediment modify the impacts of a neonicotinoid insecticide on freshwater community structure and ecosystem functioning. Science of the Total Environment, 2019, 692: 1291-1303.
Wang T., Zhang P.Y., Molinosb J., G., et al. Interactions between climate warming, herbicides, and eutrophication in the aquatic food web. Journal of Environmental Management, 2023, 345: 118753.
Comment 3: Authors should make the second paragraph of the introduction briefly and concise. The present version is too redundant.
Response 3: Thank you for your suggestion. We have reducted the second paragraph and divided into three paragraphs (Lines 65-89).
Comment 4: Line 114, 10cm deep?
Response 4: Yes, 10 cm deep, and we change the phrase of “10 cm” to “10 cm depth” (Lines 102).
Comment 5: Did authors pretreat the sediments before fill them to the mesocosm?
Response5: Yes. We pretreated the sediments, all sediments were mixed evenly, and then sieved through a 5 mm × 5 mm metal mesh to remove bulk debris, macrophyte seeds, and benthic animals. More detailed description was added in the revised manuscript (Lines 104-105).
Comment 6: In the text, author wrote the mesocosm was filled with tap water, but in the figure 1, it is water from Lake Nanhu.
Response 6: I am very sorry for the consusion. Actually, the water was added to the mesocosms by two steps. First, after the sediments were filled to the mesocosm, and the systems are basically established, the aerated tap water was added to the system 1.2 m depth, and then 10 L of original lake water from nearby Lake Nanhu was inoculated to the system. All the mesocosms were acclimated for two months before the formal experiments. We provided the information in “2.1 Experiment design” (Lines 105-113).
Comment 7: Authors collected 100ml for protozoa analysis and then concentrated before counting and identification. But they did not provide the proportion of the samples were examined. And if there were replicates please provide the information as well.
Response 7: For protozoa analysis, 100 ml water samples were collected, and were immediately fixed in Lugol’s iodine solution. After the samples were deposited for more than 48 hours, they were concentrated to 10 mL, and then finally 1 mL samples were examined. Each sample was counted twice. Detailed information was provided in the revised manuscript.(Lines 158-169)
Comment 8: Why did EP group have the highest concentration of Chl-a? Pestcide should be harmful to algae too. Authors need to explain it in the discussion.
Response 8: Thank you for your suggestion. We have added the explaination in the discussion as follows: “The high Chl-a concentration in the EP and WEP group may be because under eutrophication, the addition of N and P would proliferate the growth of phytoplankton, also because protozoan were more sensitive to imidacloprid than phytoplankton, and the removal of protozoan especially algivorous protozoan would contribute for algal proliferation. (Lines 432-436)
Comment 9: Line 352, I cannot understand “the protection of sufficient food resources”, please reword it.
Response 9: We have revised the description of “under the protection of sufficient food resources” as “when plentiful food resources were supplied” (Lines 373).
Comment 10: It is weird that in Figure 3 all error bars looks the same, Please check your data.
Response 10: Thank you for pointing out our mistake. We have checked the data and redrawn Figure 3 (Line 225).
Comment 11: Provide error bars for figure 4.
Response 11: We have added error bars for Figure 4 (Line 255).
Reviewer 2 Report
Comments and Suggestions for Authors
The report is important in terms of content. However, the statistical process remains opaque and the figures and tables are crude, making the data unreliable. The authors need to accurately describe M&M and Results.
Referring to the Yangtze River basin from the beginning of the text narrows the field to which the paper contributes. The downstream lake perspective should be clearly stated.
The wording of L85 Raptor is not appropriate. The cited reference is written in Chinese, which I have not read, but the use of Raptor could be misinterpreted as a bird species. Look for the expression from similar English language references, not this one; how about Predators?
https://www.jstage.jst.go.jp/article/jjprotozool/35/2/35_85/_pdf
L85 If you have chosen a better method compared to conventional taxonomy, choose a protozoan reference for your citation. Reference 24 is a research paper on phytoplankton.
L116 Does tap water contain any disinfectants?
  Write the capacity of the water after Water.
L118 Goldfish appear to live outdoors. If commercial goldfish are added to the aquarium, write that they are substitutes for natural crucian carp to avoid misunderstandings.
L119 If the goldfish were added to match the biomass of the lake, the weight of the goldfish should be written.
L128 Details of the computer-controlled temperature system are not clear. Does the diurnal variation in water temperature due to this system vary in the same way as the control? Was the heater put in directly or was it circulated.
At what depth were the temperature probes installed? Is there one probe per tank? Is the water in the tank also circulated? Is the average water temperature in the tank supposed to increase by 3.5°C when heated? Is it possible that only the surface is warmed? There is a risk of bias in the temperature control in the description of this paper and the references cited.
L185 It is difficult to know which statistic was used for which graph; there is no description of the X2 test.
L218 The order of the groups in Table 1 and Figure 3 differs from the order of the groups from Figure 4 onwards, so the latter should be adapted.
L253 Could not find a description in the M&M of whether protozoa abundance and biomass were really measured monthly; L158 It is mentioned that protozoa were sampled at the end of the experiment.
L257 Units of Biomass in Fig. 4 match the description in the text.
L258 Review the statistical results in Table 3.
We do not know why Table 3 does not show a significant difference between WEP populations and biomass, when here and in the text it says that WEP populations and biomass are the largest and WP biomass is significantly different.
We do not know how you performed the statistical process - did you find the mean of the monthly biomass and volume for the six iterations and apply the X2 test? Write clearly.
L279 The arrangement of panels A-D in Figure 5 is different from the order in Table 4 and needs to be aligned.
Why is this X2 test used for this statistic; ANOVA should be used.
Unclear what the squares and diamonds in the figure indicate.
L285 Table 4 Add Weiner after Shannon.
The results in Fig. 5 and Table 4 do not necessarily appear to match.
Fig. 6 Match the order of the functional groups to L83.
Match the order in Fig. 7 and Table 5.
L319 Table 6: Functional group N is not visible.
Do dominant species differ according to treatment conditions?
L327 Table 6 has no corresponding graph, is Figure 6 the corresponding graph? It is not possible to determine whether the statistical treatment is accurate based on this content.
Author Response
Comment 1: The report is important in terms of content. However, the statistical process remains opaque and the figures and tables are crude, making the data unreliable. The authors need to accurately describe M&M and Results.
Response 1: Thank you for your suggestion. We have carefully revised M&M and Results. For the part of “M&M”, we have revised the “2.3 Data processing and analysis” (Lines 171-193); for the part of results, we have redraw Figure 2 (changing the format, line 200), Figure 3 (revising error bars, line 225), Figure 4 (providing error bars, line 255), and Figure 6 (adding images of three protozoas, line 296). For the tables, the order of the treatments were changed from “C/E/P/W/EP/WE/WP/WEP” to “C/W/E/P/WE/EP/WP/WEP” for Table 1 (line 220), so that they can be corresponding to those in the other figures; “x2” was changed to “F”, and “0.0009 ***” was changed to “0.001 ***” for Table 3 (Line 261);; “x2” was changed to “F”, “Pielous” was changed to “Pielou index”, “<0.0001” was changed to “0.001” for Table 4, and the order of four diversity indexes were chaged according to Figure 5 (Line 287); “x2” was changed to “F” for Table 7 (Line 341).
Comment 2: Referring to the Yangtze River basin from the beginning of the text narrows the field to which the paper contributes. The downstream lake perspective should be clearly stated.
Response 2: Thank you for your suggestion. We removed the regional restriction on the "middle and lower reaches of the Yangtze River" in the revised manuscript, and only described as “shallow lakes”. (Lines 99)
Comment 3: The wording of L85 Raptor is not appropriate. The cited reference is written in Chinese, which I have not read, but the use of Raptor could be misinterpreted as a bird species. Look for the expression from similar English language references, not this one; how about Predators?
https://www.jstage.jst.go.jp/article/jjprotozool/35/2/35_85/_pdf
Response 3: Thank you very much for your kind suggestion. We found the original references about functional groups in protozoan (Pratt and Cairns, 1985), and both “raptor” and “predator” were used in the reference. The term “Raptor” was quoted in "Modern biomonitoring techniques using freshwater microbiota" editor by Shen et al. (1992). However, as you mentioned, “Predators” is indeed more appropriate, so all “Raptor” in the manusvript were changed to “Predators” as suggested. (Lines 77, 233, 332, 339, 348, 444, 449,457).
Pratt J R, Cairns J .Functional Groups in the Protozoa: Roles in Differing Ecosystems1,2[J].The Journal of Protozoology, 1985, 32(3):415-423.
Shen, Y.; Zhang, Z.; Gong, X.; Yan, M.; Shi, Z.; Wei, Y. Modern biomonitoring techniques using freshwater microbiota. Beijing: China Architecture & Building Press: Beijing, China, 1992, 152-175.
Comment 4: L85 If you have chosen a better method compared to conventional taxonomy, choose a protozoan reference for your citation. Reference 24 is a research paper on phytoplankton.
Response 4: Literature 24 has been deleted, and two protozoan references were cited in the revised manuscript. (Line 77)
Shen, Y.; Zhang, Z.; Gong, X.; Yan, M.; Shi, Z.; Wei, Y. Modern biomonitoring techniques using freshwater microbiota. Beijing: China Architecture & Building Press: Beijing, China, 1992, 152-175.
Pratt J R, Cairns J .Functional Groups in the Protozoa: Roles in Differing Ecosystems1,2[J].The Journal of Protozoology, 1985, 32(3):415-423.
Comment 5: L116 Does tap water contain any disinfectants?
Response 5: The tap water used in our study was aerated to remove tiny amounts of disinfectants. Besides, all the mesocosms were acclimated for two months before the formal experiments (111-112). In the revised manuscript, we changed “tap water” to “aerated tap water” (line 106).
Comment 6: Write the capacity of the water after Water.
Response 6: The capacity was about 2100L, the information was provided in the revised manuscript. (Line 106)
Comment 7: L118 Goldfish appear to live outdoors. If commercial goldfish are added to the aquarium, write that they are substitutes for natural crucian carp to avoid misunderstandings.
Response 7: It was our mistake that we didn’t describe the mesocosm systems clearly. The fish used in our study was not goldfish, but “crucian carp”, which was very common in the natural lake. Actually, the setting of the mesocosm systems was the same as the previous study of our team (Xu et al. 2023), in order to stimulate natural ecosystem in the mesocosms, turions of common submerged macrophytes were seeded in the sediment, and some fishes, shrimps and snails were introduced in each tank evenly. In order to make our manuscript concise, and not repeat the content of the reference, we re-described the mesocosm, and didn’t provide very detailed information, but cited the related reference (Line 98-113).
Xu, X., Su, G., Zhang, P., Wang, T., Zhao, K., Zhang, H., Huang, J., Wang, H., Kong, X., Xu, J., Zhang, M. Effects of multiple environmental stressors on zoobenthos communities in shallow lakes: evidence from a mesocosm experiment. Animals, 2023, 13(23), 3722.
Comment 8: L119 If the goldfish were added to match the biomass of the lake, the weight of the goldfish should be written.
Response 8: As mentioned above (Response 7), the setting of the mesocosm systems was the same as the previous study of our team (Xu et al. 2023), in order to make our manuscript concise, and not repeat the content of the reference, we re-described the mesocosm, and didn’t provide very detailed information, but cited the related reference (Line 98-113).
Comment 9: L128 Details of the computer-controlled temperature system are not clear. Does the diurnal variation in water temperature due to this system vary in the same way as the control? Was the heater put in directly or was it circulated.
Response 9: Yes, the computer-controlled temperature system was made by our team (Wang et al. 2020), and warming was achieved by using a computer-controlled system with temperature sensors (DS18B20), microcontroller (C8051F320), and a heating device, which can adjust automatically and allow real-time monitoring and recording of water temperature, so the diurnal variation of the water temperature in the systems with warming treatment vary in the same way as the control, and always 3.5 °C higher than the control. (Lines 120-127)
Moreover, temperature sensors were located 0.5 m below the water surface in both unheated and heated mesocosm, while the heater was installed 30 cm below the water surface in each heated mesocosm. (Lines 129-130).
Wang, T., Zhang, P.Y., Molinos, J.G., Xie, J.Y., Zhang, H., Wang, H., Xu, X.Q., Wang, K., Feng, M.J., Cheng, H.W., Zhang, M., Xu, J. Interactions between climate warming, herbicides, and eutrophication in the aquatic food web. Journal of Environmental Management, 2023, 345: 118753.
Comment 10: At what depth were the temperature probes installed? Is there one probe per tank? Is the water in the tank also circulated? Is the average water temperature in the tank supposed to increase by 3.5°C when heated? Is it possible that only the surface is warmed? There is a risk of bias in the temperature control in the description of this paper and the references cited.
Response 10: As we responded above (Response 9), the temperature sensors were located 0.5 m below the water surface in both unheated and heated mesocosm, and all the tanks were installed aquarium pump was installed in each mesocosm to mix and circulate water. And yes, with the use of computer-controlled temperature system, the average water temperature in the heated tank supposed to increase by 3.5°C when heated. Because the heater was installed 30 cm below the water surface, and all tanks were equipped with aquarium pumps, it is impossible that only the surface is warmed. The computer-controlled temperature system was developed by the members of our lab, and we are using the same system, so the temperature control in the description of this paper was the same as the references cited (Wang et al. 2020).
Comment 11: L185 It is difficult to know which statistic was used for which graph; there is no description of the X2 test.
Response 11: Thank you very much for your kind suggestion, we revised the part of “2.3 Data processing and analysis” to describe the statistic method used in the study. And it’s our mistake that the “X2 test” should be “Tukey's post hoc test”. (Lines 181-193).
Comment 12: L218 The order of the groups in Table 1 and Figure 3 differs from the order of the groups from Figure 4 onwards, so the latter should be adapted.
Response 12: Thank you for your suggestion. We have revised the orders of the groups in Table 1 (Line 220) and Figure 3 (Line 225) by referring Figure 4.
Comment 13: L253 Could not find a description in the M&M of whether protozoa abundance and biomass were really measured monthly; L158 It is mentioned that protozoa were sampled at the end of the experiment.
Response 13: Sorry for the confusion. Actually, we took protoan samples monthly (Line 158), however some figures or tables used the data from all months’ sampling, while some figures or tables only used the data at the end of experiment. In order to make the results more clear, the source of data were added for all figures and tables in the revised manuscript. (Line 220, 225, 256, 262, 282, 293-294, 316, 327, 342-344).
Comment 14: L257 Units of Biomass in Fig. 4 match the description in the text.
Response 14: We have revised Figure 4 to match the description in the text. (Lines 237-238)
Comment 15: L258 Review the statistical results in Table 3.
Response 15: We added the description for Table 3 in Lines 238-240.
Comment 16: We do not know why Table 3 does not show a significant difference between WEP populations and biomass, when here and in the text it says that WEP populations and biomass are the largest and WP biomass is significantly different.
Response 16: We added error bars for Figure 4, the difference between WEP and control is significant (255), however as you mentioned “Table 3 does not show a significant difference between WEP populations and biomass”, which probably caused by the interaction of other treatment. In the future, we will use more data to explore the reasons.
Comment 17: We do not know how you performed the statistical process - did you find the mean of the monthly biomass and volume for the six iterations and apply the X2 test? Write clearly.
Response 17: It was our wrong that we didn’t describe the analysis clearly. Actually, we used “tukey test” in ANOVA instead of X2 test to analysis the variance, we corrected in the revised manuscript (Line 191).
Comment 18: L279 The arrangement of panels A-D in Figure 5 is different from the order in Table 4 and needs to be aligned.
Response 18: We have the order in Table 4 as suggested. (Line 287)
Comment 19: Why is this X2 test used for this statistic; ANOVA should be used.
Response 19: It’s our mistake that we didn’t describe correctly. Actually, we used ANOVA but not X2 test to do the statistic analysis. We made the revision in the M&M (Lines 181-193).
Comment 20: Unclear what the squares and diamonds in the figure indicate.
Response 20: The squares and diamonds in the figure indicate the outliers. In the revised manuscript, we redraw Figure 5, and only used diamonds for outliers, and added decription for the diamonds (Line 281).
Comment 21: L285 Table 4 Add Weiner after Shannon.
Response 21: We have revised it as suggested. (Line 288)
Comment 22: The results in Fig. 5 and Table 4 do not necessarily appear to match.
Response 22: Yes, there were some difference between Figure 5 and Table 4, which were explained in Line 267-280.
Comment 23: Fig. 6 Match the order of the functional groups to L83.
Response 23: The order of the functional groups was based on the description in Line 83 in the original manuscript, however in our study, due to the relative abundance ratio of nonselective omnivores (N) was as low as 0.36%, the N group was not included in the functional group analysis. We provided the explaination in Lines 322-324.
Comment 24: Match the order in Fig. 7 and Table 5.
Response 24: We re-analysis the data, and added two dominant functional species, so both Fig.7 (which was changed to Fig. 6) and Table 5 were revised to match. (Lines 296, 314)
Comment 25: L319 Table 6: Functional group N is not visible.
Response 25: As mentioned in “Response 23”, Because of the small proportion of N functional group species biomass observed in the experimental system, no dominant groups were found in the statistics, and thus N is not present in Table 6.
Comment 26: Do dominant species differ according to treatment conditions?
Response 26: Yes, dominant species differ according to treatment conditions, we provided one table (Table 6, Line 316) to show the difference, and added corresponding description in the text. (Lines 308-313)
Comment 27: L327 Table 6 has no corresponding graph, is Figure 6 the corresponding graph? It is not possible to determine whether the statistical treatment is accurate based on this content.
Response 27: We added one table (Table 6, Line 316), which was corresponding to Figure 6.
Reviewer 3 Report
Comments and Suggestions for Authors
There is a need to more clearly test the experimental validity of this MS. Although the experimental content is very encouraging, it is believed that there are parts that need to be modified in the experimental treatment groups and procedures, and there are parts that need to be reconsidered in the hypothesis testing part. It is judged that it will be difficult to publish the manuscript in its current state. If possible, it would be a good idea to organize some of the material in this manuscript well and submit it in SHORT NOTE format rather than
Line 116: Why was tap water used in the mesocosm Experiments? Why was raw water from the Yangzi River not filtered and used? In addition, in figure 1, it is written that lake water used as the water for mesocosm(Lake Nanhu), but this is inconsistent with the text. If the experimental content does not match in these important areas, the reliability of the data as a MS is very.
Line 118: Are there any results on the effect of feeding mechanism on fishes in experiment 48 tanks? The purpose of including fish is not clear! The results on whether there is an effect of fish or not in this MS.
Line 192-193: The content in Figure2 is a picture that was performed 7 times each. Is the number of n=8?
Line 247: There is clear division of the feeding functional groups a protozoa in Table2, so why did the material and methods & results of this MS not include the data of bacteria and debris? In particular, the absence of biota in in situ and/or enclosure experiments is believed to have a significant impact on under or overestimation of the results, and this is believed to have a significant impact on the results of this MS.
Line 252: Figure 3 shows the change in chlorophyll concentration by control group and treatment group. Was chlorophyll concentration measured simply to show that trophic level groups are different through changes in chlorophyll concentration? It is well known that protists have high species and biomass at high trophic levels in general lake and reservoir environments. The results of this MS also cover similar content, but the originality unique to this MS is evaluated as low in the current case.
Comments on the Quality of English LanguageThere is a need to more clearly test the experimental validity of this MS. Although the experimental content is very encouraging, it is believed that there are parts that need to be modified in the experimental treatment groups and procedures, and there are parts that need to be reconsidered in the hypothesis testing part. It is judged that it will be difficult to publish the manuscript in its current state. If possible, it would be a good idea to organize some of the material in this manuscript well and submit it in SHORT NOTE format rather than
Line 116: Why was tap water used in the mesocosm Experiments? Why was raw water from the Yangzi River not filtered and used? In addition, in figure 1, it is written that lake water used as the water for mesocosm(Lake Nanhu), but this is inconsistent with the text. If the experimental content does not match in these important areas, the reliability of the data as a MS is very.
Line 118: Are there any results on the effect of feeding mechanism on fishes in experiment 48 tanks? The purpose of including fish is not clear! The results on whether there is an effect of fish or not in this MS.
Line 192-193: The content in Figure2 is a picture that was performed 7 times each. Is the number of n=8?
Line 247: There is clear division of the feeding functional groups a protozoa in Table2, so why did the material and methods & results of this MS not include the data of bacteria and debris? In particular, the absence of biota in in situ and/or enclosure experiments is believed to have a significant impact on under or overestimation of the results, and this is believed to have a significant impact on the results of this MS.
Line 252: Figure 3 shows the change in chlorophyll concentration by control group and treatment group. Was chlorophyll concentration measured simply to show that trophic level groups are different through changes in chlorophyll concentration? It is well known that protists have high species and biomass at high trophic levels in general lake and reservoir environments. The results of this MS also cover similar content, but the originality unique to this MS is evaluated as low in the current case.
Author Response
Comment 1: There is a need to more clearly test the experimental validity of this MS. Although the experimental content is very encouraging, it is believed that there are parts that need to be modified in the experimental treatment groups and procedures, and there are parts that need to be reconsidered in the hypothesis testing part. It is judged that it will be difficult to publish the manuscript in its current state. If possible, it would be a good idea to organize some of the material in this manuscript well and submit it in SHORT NOTE format rather than
Response 1: Thank you very much for your constructive suggestion. In the revised manuscript, the part of M&M was described more clearly, and all the figures and tables were re-organized, and the introduction and discussion were improved according to the reviewers. Hope the revised muanuscript could strengthen the experimental validity of this MS.
Comment 2: Line 116: Why was tap water used in the mesocosm Experiments? Why was raw water from the Yangzi River not filtered and used? In addition, in figure 1, it is written that lake water used as the water for mesocosm (Lake Nanhu), but this is inconsistent with the text. If the experimental content does not match in these important areas, the reliability of the data as a MS is very.
Response 2: It is our mistake that the methods were not described clearly. We provided the detailed process of experiment in the revised manuscript as follows (Lines 98-113):
“The bottom of each mesocosm was filled with 10 cm depth of sediment, which was collected from Lake Liangzi (N 30°11′3″, E114°37′59″) in the Yangtze River basin. All sediment was homogenized and sieved through a 5×5 mm metal mesh to remove large debris, macrophyte seeds, and snails. After that, each mesocosm system was filled to a depth of 1.2 m with about 2100 L aerated tap water first, and then ten litres of original lake water was added to the mesocosm to inoculate plankton from nearby Lake Nanhu (N 30°28′57″, E114°22′34″). An aquarium pump was installed in each mesocosm to mix and circulate water.”
As mentioned above, the aerated tap water was indeed used in the mesocosm experiments, howver, the original lake water was also inoculated to inoculate plankton in the system, and all the mesocosms were acclimated for two months before the formal experiments, so we didn’t treat the raw water from nearby lake.
Comment 3: Line 118: Are there any results on the effect of feeding mechanism on fishes in experiment 48 tanks? The purpose of including fish is not clear! The results on whether there is an effect of fish or not in this MS.
Response 3: It was our mistake that we didn’t describe setting of the mesocosm system clearly. Actually, the setting of the system was the same as the previous study of our team (Xu et al. 2023). In order to stimulate natural ecosystem in the mesocosms, we did complicated preparation for setting up the system, including seeding submerged macrophytes, introducing some fishes, shrimps and snails. Because all the setting was exactly the same as the systems in our previous study (Xu et al. 2023), we added the reference and the following description in the revised manuscript:
“The setting up of the mesocosm systems was the same as the previous stuy of our team [36].” (Lines 99-100)
“In order to stimulate natural ecosystem in the mesocosms, turions of two common submerged macrophytes were seeded in the sediment, and some fishes, shrimps and snails were introduced in each tank evenly [36]. All the mesocosms were acclimated for two months before the formal experiments.” (Lines 109-112)
For your question about the effect of feeding mechanism on fishes, because as we mentioned above, the poupose of inoculating fish is to stimulate natural ecosystem in the mesocosms, the feeding mechanism of fishes was not our focus, we didn’t analysis it in our study.
Xu X, Su G, Zhang P, Wang T, Zhao K, Zhang H, Huang J, Wang H, Kong X, Xu J, Zhang M. Effects of multiple environmental stressors on zoobenthos communities in shallow lakes: evidence from a mesocosm experiment. Animals, 2023, 13(23):3722.
Comment 4: Line 192-193: The content in Figure2 is a picture that was performed 7 times each. Is the number of n=8?
Response 4: Thank you very much for your careful reviewing. Water temperature was continuously monitored for 8 months during the whole experiment (from April to November 2021), however because the data of the last month was partially missing, water temperatures of only 7 months were used to analyze instead of 8 months. To avoid ambiguities and to further illustrate the reliability of the data, the monthly temperature variation has been changed to the daily temperature variation in this modification, as detailed in Figure 2 (Line 200).
Comment 5: Line 247: There is clear division of the feeding functional groups a protozoa in Table2, so why did the material and methods & results of this MS not include the data of bacteria and debris? In particular, the absence of biota in in situ and/or enclosure experiments is believed to have a significant impact on under or overestimation of the results, and this is believed to have a significant impact on the results of this MS.
Response 5: Thank you very much for your valuable suggestion. As you mentioned, the biota factors should have a significant impact on the the communities structure of protozoan under different environment, if our MS could provide the data of bacteria and debris, the results will be more convincing. However, we only measured one biota indicator of the chl-a consentration to express phytoplankton biomass (Line 152; Figure 3, Line 225), and found the fluctuation of algivores protozoan (which were the predominant founctional group in different treatment systems, Figre 7, Line 327), had relationship with the chl-a consentration (Line 429-436). Your suggestion inspired us to think about the ecological question from more comprehensive view, in the next study, we will develop appropriate method to measure comprehensive biota information, especially the data of bacteria and debris, to shed light on how different founctional protoan group responded to multiple environmental stress.
Comment 6: Line 252: Figure 3 shows the change in chlorophyll concentration by control group and treatment group. Was chlorophyll concentration measured simply to show that trophic level groups are different through changes in chlorophyll concentration? It is well known that protists have high species and biomass at high trophic levels in general lake and reservoir environments. The results of this MS also cover similar content, but the originality unique to this MS is evaluated as low in the current case.
Response 6: Thank you very much for your critical suggestion. In our study, the chlorophyll concentration was mainly determined to express phytoplankton biomass (Line 152), and then was used to shed light on the relationship between phytoplankton and algivores protozoan under different environmental stress.
Round 2
Reviewer 2 Report
Comments and Suggestions for Authors
The manuscript has been revised extensively on the basis of the points made and I considered it worthy of publication. If possible, to increase the value of this paper, I would like to see the responses of warming, eutrophication and pesticides on protozoa organised into a flowchart based on systems thinking. However, there were numerous careless errors. About 30 were noted, but there may be more, so authors should check.

L12: It is not common to see two Corresponding authors. Also, there are two asterisks and it is not clear whose email address it is.
L77: (R) is not appropriate as an abbreviation for Predators. If you are concerned about duplication of expression with pesticides, how about as (Pr)
L200: Add "Seasonal changes in" before "Average daily water".
L220: Rewrite Cond as Cond.
L225: Add error bars and alphabetical explanations.
L255: The lines in the Biomass column are too thin.
Add error bars and alphabetical explanation. Also add explanation of the difference between upper and lower case letters.
Rewrite Treatment on the X-axis as Treatments.
L262: (Data for this ... experiment) should be moved to footnotes. Delete the brackets
Rewrite Factors as Treatments.
L263: Write the name of the statistic here or in the text.
L281 Figure 5: Delete letters A-D in each panel if unnecessary.
Rewrite Treatment on the X-axis as Treatments.
L285: No explanation for the bars with diamonds, which are supposed to represent averages.
No explanation for BOX.
Delete the brackets for (Diamonds...outliers). Usually not used.
Diamonds showing outliers are too small to be seen as diamonds in print.
There are also diamonds in the average, which confuses the reader.
L288: The Pielou index and the Pielous index are mixed up, unify them.
L289: Factors should be rewritten as Treatments.
L289: Rewrite Factors as Treatments.
L290: Pielou index and Pielous index are mixed up, unify them.
L293: Put this explanation in the footnote above.
L314, 315: Rewrite the abbreviation for Predator from R to Pr.
L326: Rewrite Treatments on the X-axis as Treatments.
The order of the columns is different from the legend.
L326, 331: Rewrite the abbreviation for Predator from R to Pr.
L328: Delete the brackets. They are not normally used.
L341: Rewrite “Table 7” as “Table 7.”.
L341 Table 7: Rewrite Factor as Treatments.
Write the name of the statistic here or in the text. Add explanation of asterisks.
L348: Rewrite the abbreviation for Predator from R to Pr.
Author Response
Response to Reviewer 2 Comments
Comment 1: The manuscript has been revised extensively on the basis of the points made and I considered it worthy of publication. If possible, to increase the value of this paper, I would like to see the responses of warming, eutrophication and pesticides on protozoa organised into a flowchart based on systems thinking. However, there were numerous careless errors. About 30 were noted, but there may be more, so authors should check.
Response 1: We made a systematic flowchart as suggested. (Lines 360-373)
Comment 2: L12: It is not common to see two Corresponding authors. Also, there are two asterisks and it is not clear whose email address it is.
Response 2: We revised the superscripts and used “*” and “**” to show the differences. (Line 5).
Because both Yingchun Gong and Saibo Yuan provided important contribution for the manuscript, such as conceptualization, writing—original draft preparation, and writing—review and editing, we listed two corresponding authors. And we have asked advises from the editors, they told us that two corresponding authors are acceptable for the journal.
Comment 3: L77: (R) is not appropriate as an abbreviation for Predators. If you are concerned about duplication of expression with pesticides, how about as (Pr)
Response 3: We have changed “P” to “Pr”, and inserted a description that ““R” was used for “predator” in the original reference [25]” so that “Pr’ can be connected with “R” in the other reference. (Lines 77-78)
We also changed all “R” to “Pr” for Table 2. (Lines 256-258)
Comment 4: L200: Add "Seasonal changes in" before "Average daily water".
Response 4: We have revised it as suggested. (Line 201)
Comment 5: L220: Rewrite Cond as Cond.
Response 5: We have revised it as suggested (Line 221), and also changed “Cond” to “Cond.” in Line 224.
Comment 6: L225: Add error bars and alphabetical explanations.
Response 6: We have added error bars and alphabetical explanations. (Lines 226-232)
Comment 7: L255: The lines in the Biomass column are too thin.
Add error bars and alphabetical explanation. Also add explanation of the difference between upper and lower case letters.
Rewrite Treatment on the X-axis as Treatments.
Response 7: We have remade the figure, added error bars and alphabetical explanations. (Lines 259-266)
Comment 8: L262: (Data for this ... experiment) should be moved to footnotes. Delete the brackets
Rewrite Factors as Treatments.
Response 8: We have revised it as suggested. (Lines 268-274)
Comment 9: L263: Write the name of the statistic here or in the text.
Response 9: We provided the name of the statistic as suggested (Line 271)
Comment 10: L281 Figure 5: Delete letters A-D in each panel if unnecessary.
Rewrite Treatment on the X-axis as Treatments.
Response 10: We revised the figure as suggested. (Line 290)
Comment 11: L285: No explanation for the bars with diamonds, which are supposed to represent averages.
No explanation for BOX.
Delete the brackets for (Diamonds...outliers). Usually not used.
Diamonds showing outliers are too small to be seen as diamonds in print.
There are also diamonds in the average, which confuses the reader.
Response 11: We revised the figure, and added the description for the BOX and diamonds. (295-296)
Comment 12: L288: The Pielou index and the Pielous index are mixed up, unify them.
Response 12: We have checked through the manuscript, and changed “Pielous” to “Pielou” (L390), and changed “Pielou” to “Pielou’ s (Line 171, 279).
Comment 13: L289: Factors should be rewritten as Treatments.
Response 13: We have revised it as suggested (Line 299).
Comment 14: L289: Rewrite Factors as Treatments.
Response 14: We have revised it as suggested (Line 299).
Comment 15: L290: Pielou index and Pielous index are mixed up, unify them.
Response 15: We have checked through the manuscript, and changed “Pielous” to “Pielou” (L414), and changed “Pielou” to “Pielou’ s (Line 172 287).
Comment 16: L293: Put this explanation in the footnote above.
Response 16: We have revised it as suggested (Line 300-310).
Comment 17: L314, 315: Rewrite the abbreviation for Predator from R to Pr.
Response 17: We have revised it as suggested (Lines 325, 326).
Comment 18: L326: Rewrite Treatments on the X-axis as Treatments.
The order of the columns is different from the legend.
Response 18: We have revised the figure as suggested. (Line 337)
Comment 19: L326, 331: Rewrite the abbreviation for Predator from R to Pr.
Response 19: We have revised it as suggested (Line 341, 350, 357).
Comment 20: L328: Delete the brackets. They are not normally used.
Response 20: The brackets were deleted.
Comment 21: L341: Rewrite “Table 7” as “Table 7.”.
Response 21: We have revised it as suggested (Line 350), and revised all the similar mistakes through the manuscript. (Lines 98, 145, 171, 196, 233, 306, 325, 327, 361, 440, 492)
Comment 22: L341 Table 7: Rewrite Factor as Treatments.
Write the name of the statistic here or in the text. Add explanation of asterisks.
Response 22: We have revised it as suggested (Line 355, 356).
Comment 23: L348: Rewrite the abbreviation for Predator from R to Pr.
Response 23: We have revised it as suggested (Line 350).
